# Screening and Structure–Activity Relationship for Selective and Potent Anti-Melanogenesis Agents Derived from Species of Mulberry (Genus *Morus*)

**DOI:** 10.3390/molecules27249011

**Published:** 2022-12-17

**Authors:** Anna Gryn-Rynko, Beata Sperkowska, Michał S. Majewski

**Affiliations:** 1Department of Pharmacology and Toxicology, Faculty of Medicine, University of Warmia and Mazury, Warszawska 30 Street, 10-082 Olsztyn, Poland; 2Department of Toxicology and Bromatology, Faculty of Pharmacy, Ludwik Rydygier Collegium Medicum in Bydgoszcz, Nicolaus Copernicus University in Toruń, A. Jurasza 2 Street, 85-089 Bydgoszcz, Poland

**Keywords:** hyperpigmentation, tyrosinase inhibitors, tyrosine, genus *Morus*

## Abstract

Tyrosinase is a multifunctional, copper-containing and rate-limiting oxidase that catalyses crucial steps in the melanogenesis pathway and is responsible for skin-pigmentation abnormalities in mammals. Numerous tyrosinase inhibitors derived from natural and synthetic sources have been identified as an objective for the development of anti-melanogenesis agents. However, due to side effects and lack of expected efficiency, only a small percentage of them are used for medical and cosmetic purposes. This critical review focuses on searching for novel active substances and recently discovered plant-derived anti-tyrosinase inhibitors from the *Morus* genus (*Moraceae* family). A detailed analysis of their structure–activity relationships is discussed. The information contained in this article is crucial for the cosmetics and medical industries, in order to show new directions for the effective search for natural anti-melanogenesis products (with satisfactory efficiency and safety) to treat and cure hyperpigmentation.

## 1. Introduction

The colour of human skin, hair and eyes is determined by the amount and type of melanin pigment produced in melanosomes by cutaneous and follicular melanocytes, located in the basal layer of the skin epidermis [1,2]. Through the dendritic process, melanocytes interact with approximately 36 adjacent keratinocytes in order to transport mature melanosomes and protect the skin from photo-induced carcinogenesis [2,3]. Melanocytes in mammals produce two chemically distinct types of melanin: the yellow to reddish pheomelanin (1A) and the black to brown eumelanin (1B) [4]. An essential role in the melanogenesis pathway (Figure 1) is played by the tyrosinase gene family protein, i.e., tyrosinase (Tyr), tyrosinase-related protein-1 (TRP-1) and tyrosinase-related protein-2 (TRP-2) [3,4], and their transcription factors, such as microphthalmia-associated transcription factor (MITF), cAMP response element binding protein (CREB) and extracellular-regulated kinase (ERK) [5]. Among oxidases, Tyr is a key, rate-limiting melanogenic enzyme that catalyses two crucial initial steps in melanogenesis, such as the hydroxylation of the monophenol L-tyrosine to the o-diphenol 3,4-dihydroxyphenylalanine (L-DOPA) (monophenolase or cresolase activity) and the oxidation of L-DOPA to dopaquinone (diphenolase or catecholase activity) [2,3]. Essential for the catalytic activity of tyrosinase are two copper atoms (CuA and CuB), located in the active site of the enzyme, surrounded by a bundle of four helices and coordinated by six histidine residues [6]. During the catalytic reaction, the active centre of tyrosinase exists in three redox forms: oxy-tyrosinase, met-tyrosinase and deoxy-tyrosinase [6]. Tyrosinase is widely distributed and can be isolated and purified from various sources, including bacteria, fungi, plants and mammals [7]. The best-described tyrosinase is derived from bacteria Streptomyces glaucescens and the fungi *Neurospora crassa* and *Agaricus bisporus* [4,8,9].

Although melanin (**1**) has a mainly photoprotective function in human skin. Overproduction, accumulation and abnormal distribution of this natural pigment may contribute to the occurrence of irregular skin hyperpigmentary disorders [3,4,8,10]. Exposure to certain drugs and chemicals as well as vitamin deficiency and the existence of metabolic disorders are also linked to local hyperpigmentation [8,11]. Furthermore, according to recent publications, many melanogenesis disorders are reported to be associated with neurodegenerative diseases, including Alzheimer’s and Parkinson’s [12,13]. Therefore, there is a huge need for searching for novel formulations to develop low-cost therapies through unknown to data ingredients that can help to treat and prevent hyperpigmentary disorders.

Since a tyrosinase-catalysed reaction is highly associated with hyperpigmentation, discovering tyrosinase inhibitors is of great importance in medicinal, cosmetic and cosmeceutical products for the treatment and prevention of pigmentation disorders [14]. Unfortunately, the capacity of commercially available tyrosinase inhibitors as whitening agents is not completely satisfactory [14,15,16,17,18,19,20]. Although they can efficiently inhibit tyrosinase activity, their exploitation for this purpose has been limited due to the side effects and the lack of safety regulations governing their use [20]. For example, hydroquinone (**2**), one of the most potent and frequently prescribed tyrosinase inhibitors, causes significant pathological conditions, such as skin irritation, contact dermatitis and ochronosis, and might have cytotoxic effects on melanocytes [14,15,16,17,18,19,20,21]. Because of the risks of adverse effects, the European Committee banned hydroquinone, its derivative formulation has been withdrawn from cosmetics and it is only available through prescription [1]. Unfortunately, none of the alternatives to hydroquinone (**2**), such as arbutin (**3**) or kojic acid (**4**), are offered as a safe and effective replacement (Figure 2) [14,16,17,18,19,20].

Recently, the plant extracts and isolated compounds from natural sources have been extensively utilized for searching new, safe and effective products with antityrosinase activity, as a novel approach for the treatment of pigmentary disorders [14,22,23,24]. Parts of the species of the *Morus* genus are well known for their multidirectional therapeutic uses, such as in tyrosinase inhibition [25,26,27,28,29,30,31,32,33]. Thus, the aim of the present review is to update the recent knowledge about tyrosinase inhibitors derived from the family *Moraceae*, genus *Morus* [25], which is one of the richest in bioactive compounds. Moreover, the structure–activity relationship is also discussed, wherever possible. The information offered here is designed to develop new melanogenesis inhibitors of adequate efficacy and safety that might inspire further examination of topical drugs or cosmetic formulations for the treatment of hyperpigmentary disorders.

## 2. Polyphenols

Nowadays, polyphenolic compounds are the largest and the main groups responsible for tyrosinase inhibitor activity [34], probably due to their structural similarity to tyrosine [35]. Polyphenols represent a group of naturally occurring compounds, from simple molecules to complex structures, containing multiple phenolic functionalities and extensively distributed in nature [36]. Due to their beneficial biological properties, they are used in numerous sectors in the industry, including their use as ingredients in cosmetic and cosmeceutical products. Polyphenols are chemical compounds with more than one hydroxyl functional group attached to an aromatic ring [37]. The presence and position of additional constituents determine the activity of polyphenols as tyrosinase inhibitors [36]. Polyphenols can be subdivided into two major subclasses: flavonoids (flavones, flavonols, flavanones, isoflavones, flavanonones, flavans, flavanols, chalcones, dihydrochalcones, flavan-3,4-diols, anthocyanins) and non-flavonoids (phenolic acids, stilbenes, coumarins and tannins) [38,39]. In this review, a detailed summary and discussion of the scientific results on the discovery of tyrosinase inhibitors, naturally occurring in the *Moracae* family, *Morus* genus, will be carried out according to the chemical structures of polyphenols.

### 2.1. Flavones

Many flavones have been isolated from the Moraceae family, but only a few of them are worthy of particular attention as potent melanogenesis inhibitors (Figure 3). One of them is norartocarpetin (**5**) (5,7,2′,4′-tetrahydroxyflavone), isolated from the roots and stem barks of *M. lhou*, which showed it is 13.58- (IC_50_ = 1.2 μM) and 10.42-fold (IC_50_ = 1.2 μM) more effective toward competitive monophenolase (K_i_ = 0.61 µM) activity of mushroom tyrosinase than kojic acid (**4**) [40,41]. Similar results presented by Yang et al. prove that norartocarpetin (**5**), determined in *M. alba* leaves, is 192.96-times stronger as a tyrosinase inhibitor than kojic acid (**4**) [26]. Norartocarpetin (**5**) effected slow-binding enzyme isomerization [40]. Furthermore, it was noted that the position and number of the hydroxyl group in the B ring of the norartocarpetin structure could strongly affect the inhibitory strength. An unsubstituted resorcinol group, especially the 2′ and 4′-OH, deserves special attention because this structure is associated with strong inhibitory activity [26].

Further, also noteworthy is that moracenin D (**6**) (IC_50_ = 4.61 μM), identified from the roots of *M. australis*, was proved to possess an anti-tyrosinase activity eleven-times stronger than kojic acid (**4**). The inhibitory strength of the other two isolated compounds, kuwanon G (**7**) (IC_50_ > 200 μM) and kuwanon N (**8**) (IC_50_ = 78.95 μM), was found to be poorly active toward mushroom tyrosinase compared with kojic acid (**4**). Considering the structure–activity relationship, it can be assumed that the presence of an isoprenyl group in position 3 of the flavonoid skeleton greatly reduces the inhibitory activity (as demonstrated in kuwanon G (**7**) and N (**8**)). According to Dej-adisai et al., the substitution at position C-3 is critical for the tyrosinase inhibitory activity of the flavone group [42]. However, the replacement of the isoprenyl group by an isoprenyl alcohol group results in an increase in tyrosinase activity (as demonstrated in moracenin D (**6**)) [27].

Among the newly discovered tyrosinase inhibitors, isolated from the roots of *M. nigra*, kuwanon H (**9**) (IC_50_ =10.34 μM) and 5′-geranyl-5,7,2′,4′-tetrahydroxyflavone (**10**) (IC_50_ = 37.09 μM) deserve particular attention. They showed 4.54- and 1.27-fold higher antimonophenolase activity than the kojic acid (**4**) (IC_50_ = 46.95 μM). It was noted that the presence of an isoprenyl/ geranyl group at different positions in compound (**10**), especially the B ring of the flavonoid skeleton, affects tyrosinase inhibitory capacity [28]. The other purified flavones, such as kuwanon G (**7**) (IC_50_ > 200 μM), cudraflavone C (**11**) (IC_50_ > 200 μM) and cudraflavone B (**12**), present weak tyrosinase inhibitory activity [28]. The authors suggest that much stronger tyrosinase inhibitory activity of kuwanon H than kuwanon G was probably due to the presence of an additional isoprenyl group at the 24 position on the E ring [28].

### 2.2. Flavanones and Flavonones

One of the promising compounds obtained from the cortex of *M. bombycis* is 7,2′,4′-trihydroxyflavanone (**13**), which was proved to be 21.56-fold more potent than kojic acid (**4**) in the monophenolase inhibitory activity of mushroom tyrosinase [29].

An interesting compound, purified from the roots of *M. nigra*, is steppogenin-7-O-β-D-glucoside (**14**), which presented the strongest monophenolase properties (IC_50_ =5.99 μM) from four analysed flavonoids and was 7.84-fold stronger than kojic acid (**4**) toward mushroom tyrosinase [28]. Because steppogenin-7,40-di-O-β-D-glucoside was more than 50-fold weaker than the steppogenin-7-O-β-D-glucoside, it can be concluded that glycosidation of hydroxyl groups, especially at the 4′ position, negatively influences the tyrosinase inhibitory activity of flavonones [29].

Free steppogenin (**15**) (5,7,2′,4′-tetrahydroxyflavanone) was also determined as a highly competitive monophenolase inhibitor (Ki = 0.7 μM, IC_50_ = 1.3 μM) present in the roots of *M. Ihou* and the twigs of *M. alba* (IC_50_ = 0.98 μM), with tyrosinase activity 12.54- and 59.49-fold stronger than kojic acid (**4**) (IC_50_ = 16.3 μM, 58.30 μM, respectively) [40,43]. The results also indicated that steppogenin (**15**) inhibits mushroom tyrosinase by simple and reversible slow binding when L-tyrosine was used as a substrate [40]. The inhibitory effect of steppogenin on tyrosinase activity is time dependent [40].

According to Kim et al., flavonoid derivatives containing sugar (as demonstrated by steppogenin-7-O-β-D-glucoside (**14**)) show less tyrosinase inhibitory activity than their aglycones (as demonstrated by steppogenin), because D-glucose bulky conformation hinders their approach to the copper-containing active side in the enzyme [44,45].

It is noteworthy that the chemical structure of steppogenin (**15**), a flavanone, is very congruent to that of norartocarpetin (**5**), a flavone with the same four substituted hydroxyl groups [36]. It is reasonable that those two flavonoids, (**5**) and (**15**), possess similar inhibitory capacity and the same slow-binding and competitive inhibition mode (K_i_ = 0.7 μM (**14**) and K_i_ = 0.61 μM (**5**), respectively) against L-Tyrosine. It is also important to note that those two potent tyrosinase inhibitors contain 4-resorcinol in the B ring [36,40].

Another research group, led by Zheng et al., carried out a lot of auspicious work for the three identified active tyrosinase inhibitors from *M. australis* roots, including sanggenon T (**16**) (IC_50_ = 1.20 μM), kuwanon O (**17**) (IC_50_ =1.81 μM) and kuwanon L (**18**) (IC_50_ = 58.82 μM). The two first flavonones showed excellent antimonophenolase activity toward mushroom tyrosinase, 42- and 27.86-fold stronger than kojic acid (**4**), while the third (**18**) constituent possesses inhibitory activity equal to that presented by kojic acid (**4**). The authors suggest that the substitution of an isoprenyl group at the 10′′ position of kuwanon O (**17**) is responsible for much stronger tyrosinase inhibitory properties than what is presented by kuwanon L (**18**) [27].

Recently, Kim and co-workers conducted a lot of insightful research analysis to investigate the inhibition mechanism of a large group of flavonoids with hydroxyl moiety at various positions a using fluorescence-quenching spectroscopy method [44]. The results of their study showed that copper-containing oxidase is mostly quenched by the hydroxyl groups of A and B rings on either side of the flavonoids (C6 to C8 and C2′ to C4′) [36,43,44]. In contrast, an increase in the number of hydroxyl groups on the ketone side (C-5, C-6, C-5′ C-6′) reduces the activity of the inhibitors [46]. That explains why the modifications of the number and location of hydroxyl substituents caused changes in the tyrosinase inhibitory activities [43,46]. According to Gębalski et al., the location of the hydroxyl groups plays a fundamental role and is even more important than their number. Moreover, an increase in the number of hydroxyl groups in flavonoids diminishes their inhibitory activity against tyrosinase [46]. Jeong et al. found that the addition of prenyl groups or the replacement of the 4′hydroxyl moiety in the A and/or B ring decreases the inhibitory strength [47]. The molecular docking analysis also supports the hypothesis that the dicopper catalytic site of tyrosinase is a preferential binding site for flavonoids, thereby explaining the variety of tyrosinase inhibitory activities by flavonoids substituted in several hydroxyl groups [44]. Their data may also clarify the poor and moderate inhibitory capacity of tyrosinase by most of their type of flavones, flavanones and flavonones present in the *Morus* genus [36].

Structures of flavanones and flavonones are presented in Figure 4.

### 2.3. Chalcones

Chalcones structurally represent open-chain flavonoids, in which the two aromatic rings are joined by a three-carbon α, β-unsaturated carbonyl system (1,3-diphenyl-2-propen-1-one) [48]. Some natural chalcones exhibited very potent tyrosinase inhibitory activity (Figure 5), such as morachalcone A (**19**) (2,4,2′,4′-tetrahydroxy-3-(3-methyl-2-butenyl)-chalcone, TMBC) and 2,4,2′,4′-tetrahydroxychalcone (**20**). Those two chalcones derivatives, isolated from the roots of *M. nigra*, showed the most potent monophenolase inhibitory activity of mushroom tyrosinase from 27 investigated compounds by Zheng et al. [28]. Among them, 2,4,2′,4′-tetrahydroxychalcone (**20**) (IC_50_ = 0.062 μM) was 757.26-fold more effective than kojic acid (**4**), while TMBC (**19**) (IC_50_ = −0.14 μM) was proved to inhibit tyrosinase capacity 335.36-fold stronger than the control compound [28].

These results are in accord with those presented by Kang and co-workers, where 2,4,2′,4′-tetrahydroxychalcone (**20**), isolated from the cortex of *M. bombycis*, was found to be 115.90- and 3.8-fold more potent than kojic acid (**4**) and even oxyresveratrol**,** respectively [29].

The authors’ results emphasize the importance of the two functional resorcinol units in increasing potent tyrosinase inhibitory activity [29].

Similarly, TMBC (**19**), isolated from the stems of *M. nigra*, was identified as a classical, dose-dependent, competitive tyrosinase inhibitor, which was 26.19-times the activity of kojic aid against monophenolase activity of mushroom tyrosinase [49]. In agreement with those data is Zhang et al., who found that 2,4,2′,4′-tetrahydroxychalcone (**20**) is the most potent anti-melanogenesis inhibitor from the twigs of *M. alba* (IC_50_ = 0.07 μM) followed by morachalcone A (**19**) (IC_50_ = 0.08 μM) [43]. Takahashi et al. reported that 2,4,2′,4′-tetrahydroxychalcone (**20**) (IC_50_ = 0.21 μM) and TMBC (**19**) (IC_50_ = 0.82 μM), isolated from *M. australis* steams, showed significantly increased cellular tyrosinase activity, 780.95- and 200-fold stronger than arbutin (**3**), respectively [50]. Moreover, Takahashi and co-workers recently successfully synthesized one new active tyrosinase inhibitor 3′-[(E)-4′’-hydroxymethyl-2″-butenyl]-2,4,2′,4′-tetrahydroxychalcone (**21**) (IC_50_ = 0.17 μM), which was found to be 964.70-fold more active than arbutin [50]. Furthermore, the conducted studies showed that the three above-mentioned hydroxychalcones, (**20**), (**19**) and (**21**), efficiently suppressed cellular tyrosinase activity and melanin content in B16 murine melanoma cells (B16 cells), with little or no cytotoxicity [35,47,49,50]. Moreover, the inhibition strength of melanin synthesis in B16 cells of the three chalcones 2,4,2′,4′-tetrahydroxychalcone (**20**), TMBC (**19**) and 3′-[(E)-4′’-hydroxymethyl-2′’-butenyl]-2,4,2′,4′-tetrahydroxychalcone (**21**) was more than 100-fold greater than the one of arbutin (**3**) (IC_50_ = 5.0 μM, 3.8 μM, 4.0 μM) [50]. The structure–activity relationship of chalcones discussed here also supported the present findings that the 4-resorcinol moiety (2,4-dihydroxyl groups in the aromatic rings) is essential for exerting potent tyrosinase inhibition activity [35,36,50], especially the 4-hydroxyl group in the B ring, due to the close similarity to the amino acid L-tyrosine molecule [35,36,46] and the strong chelating properties of the copper ions, which act as the active site of tyrosinase with the 2,4-hydroxyl resorcinol [50]. The position of the hydroxyl groups attached to the A and B aromatic rings is of major importance, while hydroxylation on ring B contributes markedly more to inhibition than when it is on ring A [35].

The presence of a sugar residue at the 4′-OH position and the absence of the 3′-OH group in the B ring of chalcones resulted in reduced inhibitory activity [46]. According to Nerya et al., when one or two hydroxyls are present only in ring A of the chalcone moiety or when no hydroxyl groups have been attached to the chalcone skeleton, the other constituents of its molecule are practically inactive [35]. Based on these findings, including high monophenolase activity and effective inhibition of melanogenesis in B16 cells without discernible cytotoxic effects, it can be concluded that the hydroxychalcones could be excellent and very promising ingredients in the cosmetic industry and can open a new direction for the treatment of hyperpigmentation disorders [35,36,49,50,51].

### 2.4. Stilbenes

Stilbene consists of an ethene double bond substituted with a benzyl ring on both carbon atoms of the double bond [36]. One of the glycosylated stilbenes, widely employed as an effective ingredient in skin-lightening cosmetics [52,53,54,55,56], is mulberroside A (**22**) (oxyresveratrol-4-O-ß-D-glucopyranosol-3′-O-ß-D-glucopyranoside), isolated for the first time in 1986 from *Morus Ihou Koidz* root and presented in Figure 6 [52].

The tyrosinase activity of naturally occurring stilbene was determined in the branches (*M. multicaulis* Perr.) and roots (*M. alba* Linn.) of *Morus* genus [52,57]; the conducted studies, however, show its presence in the bark (*M. alba*, *M. atropurpurea* Roxb., *M. latifolia*), branches (*M. atropurpurea* Roxb., *M. multicaulis* Perr*., M. tiliaefolia* Makino*, M. bombycis* Koidz.*, M. alba* Linn.*, M. wittiorum* Hand-Mazz), pith (*M. atropurpurea* Roxb.), roots (*M. alba*, *M. atropurpurea* Roxb., *M. latifolia*), steam (*M. alba, Morus atropurpurea Roxb., M. latifolia*) and tuber (*M. atropurpurea* Roxb.) [52,57,58,59].

Mulberroside A (**22**) (mulA) acts as a competitive inhibitor of diphenolase activity (K_i_ = 4.36 µmol/L), while the monophenolase activity showed mixed type 1 inhibition (K_i_ = 0.385 µmol/L) [52].

It has also been proven that this inhibitor reduces depigmentation, decreases melanin content and inhibits melanogenesis through the down-regulation of gene expression, including MITF, Tyr, TRP-1 and TRP-2, more effectively than arbutin (**3**), a high skin-lightening and depigmenting agent [55,60].

However, the active tyrosinase inhibitor (**22**) was reported to possess weak oral bioavailability (<1%) in the rat [57,61], while the absorption ratio of the inhibitor derivatives was estimated at about 50%. Experiments conducted by Qiu et al. [61] and Zhaxi’s group [57] identified three active metabolites of mulA (**22**): oxyresveratrol (**23**) (OXY) and oxy conjugates (monoglycosides, glucuronidate and/or sulfate) in rat intestinal track, faeces, urine, plasma and bile after oral application of either mulA (**22**) or aqueous extract of *Mori* cortex (dry root bark of *Morus alba*). An in vitro study by Mei and co-workers [62] on Caco-2 cells demonstrated a rapid hydrolysis of mulA (**22**) to aglycone oxy in intestinal bacterial deglycosylation and an extensive hepatic conjugation of OXY (**23**) to two derivatives: major glucuronidation and minor sulfation products.

It has been suggested that mulA (**22**) expresses its pharmacological effects after being transformed to oxyresveratrol (**23**) (2,4,3′,5′-tetrahydroxy-trans-stilbene) [55,62,63].

OXY (**23**), as one of the most potent tyrosinase inhibitors [27,28,29,43,53,55], was found in the bark (*M. alba*, *M. atropurpurea* Roxb., *M. latifolia*), branches (*M. multicaulis* Perr.), fruits (*M. atropurpurea* Roxb., *M. multicaulis* Perr., *M. nigra* Linn., *M. cathayana* Hemsl., *M. alba* Linn., *M. laevigata* Wall., *M. australis* Poir.), leaves (*M. alba* Linn., *M. latifolia, M. atropurpurea* Roxb., *M. multicaulis* Perr., *M. nigra* Linn., *M. cathayana* Hemsl., *M. laevigata* Wall., *M. australis* Poir.), pith (*M*. *atropurpurea* Roxb.), roots (*M. alba*, *M. atropurpurea* Roxb., *M. latifolia, M. nigra*), steam (*M*. *alba, M. atropurpurea* Roxb.*, M. latifolia, M. rubra*), twigs (*M. alba*) and tuber (*M*. *atropurpurea* Roxb.) of the *Moraceae* family [43,52,57,58,64].

The main metabolites of OXY (**23**) are monoglucuronided oxyresveratrol and monosulfated oxyresveratrol [65]. The regulation of melanin production is one of the most important fields of oxyresveratrol activity. According to Wang et al., OXY (**23**) presents dual effects: competitive–non-competitive on monophenolase and non-competitive on diphenolase activity of mushroom tyrosinase [52]. Moreover, OXY (**23**) significantly down-regulated the expression of Tyr, TRP-1, TRP-2 and MITF genes in the epidermis of UVB-irradiated brown guinea pig skins [55] and, in particular, treated the brown guinea pig skin samples with oxyresveratrol (**23**) and oxyresveratrol-3-O-glucoside (**24**), causing a reduction in tyrosinase expression levels by 55% and 40%. The authors found that application with 5% OXY (**23**) inhibited the melanogenesis (reduction to 46%) greater than tyrosinase expression and activity (reduction to 58%) [55]. OXY (**23**) appeared to inhibit human tyrosinase activity stronger than murine or mushroom Tyrs [66]. It is also worth noting that OXY (**23**) contains the conformation of 4-resorcinol moiety as in both rings A and B of stilbenes structures. Several studies have clearly demonstrated that the aglycone of mulA (**22**) possesses much greater bioactive properties than the parental compound [28,52,53]. According to Kim et al., [53] OXY (**21**), obtained from roots of *M. alba*, exerted 109.39-, 43.06- and 1503.06-fold greater monophenolase inhibitory activity than parental compound—mulA (**22**), kojic acid (**4**) and arbutin (**3**), respectively [53], while in the roots of *M. australis,* the OXY (**23**) was found to inhibit 17.69-times stronger than kojic acid (**4**) [27].

Zheng et al. [28] suggested that the derivatives of OXY (**23**), such as oxyresveratrol-3′-O-β-D-glucopyranoside (**24**) (IC_50_ = 29.75 µM) and oxyresveratrol-2-O-β-D-glucopyranoside (**25**) (IC_50_ = 1.64 µM), isolated from the roots of *M. nigra,* present much better tyrosinase inhibitory activity than mulA (**22**) (IC_50_ > 200 µM). The authors reported that the glycosidation of mulberroside A (**22**) plays a major role and resulted in enhanced inhibition of melanogenesis [53,55]. The presence of the bulky and hydrophilic D-galactopyranosyl moiety in mulA (**22**) interferes with the entrance of the molecule into the active site of the enzyme, thus, reducing its inhibitory activity [36]. Researchers hypothesize that glycosylation at the 2 or 4 positions of (**22**) suppresses the activity more effectively than at the 3′ or 5′ positions [28]. Furthermore, it was reported that a free hydroxyl group at the 4 position of the resorcinol structure of resorcinol undoubtedly plays an important function in mediating inhibitory activity based on the observation that glycosidation at this site led to substantially weakened activity [28]. In addition, Liu et al. [32] reported that the compound oxyresveratrol-3′-O-β-D-glucopyranoside (**24**) had a competitive type of tyrosinase inhibition, indicating that this structure is related to the substrate and can competitively bind to the active centre of the enzyme, while the inhibition type of oxyresveratrol-2-O-β-D-glucopyranoside (**25**) on tyrosinase belongs to the mixed mode, in which the inhibitor will not only bind the substrate with the active centre of the enzyme but also bind to the enzyme substrate complex.

### 2.5. Arylobenzofurans

2-arylbenzofuran derivatives are the active components isolated and identified from *M. alba*, *M. australis*, *M. nigra*, *M. lhou* and *M. macroura* [8,26,27,29,40,67,68,69,70]. However, researchers indicate that only a few of them possess tyrosinase inhibition activities, such as moracin C (**26**), D (**27**), M (**28**), N (**29**) and O (**30**) [26,27,28,29,40,47] so far (Figure 7).

According to Zheng and co-workers, the most potent inhibitor from isoprenyl-substituted 2-arylbenzofuran is moracin N (**29**) (2-(3,5-Dihydroxyphenyl)-6-hydroxy-5-prenylbenzofuran) isolated from roots of *M. nigra* (IC_50_ = 30.52 μM), which exhibited 1.54-, 3.06- and 3.65-fold stronger inhibition properties than kojic acid (**4**), moracin O (**30**) and moracin C (**26**), respectively [28]. Those findings were also consistent with those presented by Yang et al., who found that moracin N (**30**) (IC_50_ = 0.924 μM), obtained from the leaves of *M. alba,* is 2.46-, 12.99- and 17.21-fold stronger than moracin C (**26**), D (**27**) and kojic acid (**4**), respectively [28]. In moracin O, the isoprenyl group forms a five-membered ring with the hydroxyl group at the 6 position, whereas in moracin N (**30**), the isoprenyl group remains intact, which might contribute to its higher tyrosinase inhibitory activity relative to moracin O (**30**) [28,70]. Different positions of the isoprenyl group also result in different tyrosinase inhibitory properties, as in the case of moracin C (**26**) and moracin N (**29**) [28].

Another 2-arylbenzofuran, moracin M (**28**) (Veraphenol, 6,3′,5′-Trihydroxy-2-phenylbenzofuran), which acts competitively on monophenolase (K_i_ = 7.4 µM) and diphenolase (K_i_ = 64.6 µM) activity of mushroom tyrosinase, isolated from the roots of *M. Ihou*, possesses 2.2- and 21.6-fold stronger mushroom tyrosinase inhibitory activity than kojic acid (**4**) and moracin N (**29**), respectively [40]. The results indicate that an addition of hydroxyl or prenyl group (as demonstrated by moracin N (**29**)) to the benzofuran skeleton decreased the tyrosinase inhibitory activity [70]. In agreement with those findings is Zhang et al., who concluded that moracin M, isolated from twigs of *M. alba*, is one of the main compounds responsible for the potent tyrosinase inhibitory activity (IC_50_ = 8.0 μM), much stronger than that of the positive control kojic acid (**4**) (IC_50_ = 58.30 μM) [43].

On the other hand, Zheng and co-workers found that moracin M (**28**) from the roots of *M. australis* possesses 51% lower tyrosinase activity than kojic acid (**4**) [27].

Apart from moracin M (**28**), another 2-arylbenzofuran derivative was identified as a tyrosinase inhibitor. Mulberroside F (**31**) (moracin M-6, 3′-di-O-β-D-glucopyranoside) purified from the leaves of *M. alba,* as compared with kojic acid (**4**), possesses 4.48-fold greater monophenolase activity of mushroom tyrosinase, while mammalian antityrosinase activity was suppressed to 50% at a concentration of 68.3 µg/mL. Further, compound (**31**) is more effective in inhibiting melanin production in cultured melan-a cells [68]. Diels-Alder type of 2-arylbenzofuran–mulberrofuran G (**32**) (Albanol A) isolated from roots of *M. nigra* show 10-fold higher inhibitor properties than mulberrofuran J (**33**) [28], while from the roots of *M*. *austarlis,* displayed 3.44-fold inhibitory activity of kojic acid (**4**) against mushroom tyrosinase [27]. Moracinoside M (**34**) (2S,3R,4S,5R)-2-[3-[5-[(3,3-dimethyloxiran-2-yl)methyl]-6-hydroxy-1-benzofuran-2-yl]-5-hydroxyphenoxy]oxane-3,4,5-triol), isolated from the cortex of *M. bombycis,* displayed 1.84-times more activity than that of kojic acid (**4**) [29].

Generally, the 2-arylbenzofuran derivatives obtained from *Moraceae* had lower tyrosinase inhibitory properties than the stilbene compounds, suggesting that part of the tyrosinase inhibitory functionality is lost upon formation of the five-membered ring in the 2-arylbenzofuran derivatives [28,71]. Furthermore, the resorcinol moiety of 2-arylbenzofuran derivatives may result in decreased tyrosinase inhibition. Further, substitutions, including glucose groups at the C-3 position of the C ring, effectively attenuate the inhibition of tyrosinase activity [71].

## 3. Application in Cosmetic and Pharmaceutical Industry

Various parts of plants from genus *Morus*, including root bark, branches, leaves and fruits, are used as a cosmetic ingredient in most Asian countries [71,72,73,74,75,76,77,78]. In a study conducted by Budiman et al. [72], a peel-off mask gel for topical treatment, containing 1.5% of *Morus nigra* leaves extract, 15% of polyvinyl alcohol and 0.5% of carbomer, was evaluate, which can inhibit tyrosinase enzyme activity. The results showed that black mulberry leaves extract had lower activity of tyrosinase (IC_50_ = 511.91 μg/mL) than kojic acid (IC_50_ = 47.19 μg/mL) but can still be a potential material as a tyrosinase inhibitor. The authors suggest that the formulation can be used as a safe cosmetic preventing hyperpigmentation [72]. The gel of black mulberry fruit extract was also categorized as a safe topical formulation [73]. Akhtar et al. [74] reported that cream with mulberry extract can be a good choice to reduce skin disorders. A formulated, stable O/W cream, containing 4% concentrated ethanolic extract of *Morus alba* L. root bark extract, significantly decreased the melanin and erythema contents on human skin without causing irritation [74]. Zakeri et al. [76] prepared a manufactured 3% cream containing crude 70% ethanolic extract of *Morus alba* L. leaves. All the tested parameters, such as physicochemical, total phenolic content and microbial tests, were satisfactory. According to the authors, cream is a very promising formulation for in vivo use as a skin-lightening agent. Similar results presented by Abhijith et al. [77] prove that herbal cold cream using *Morus alba* may prevent hyperpigmentation or patches on the dark skin and naturally lighten the skin tone. It can also protect the skin from environmental factors, hydrate, nourish and help to prevent premature aging. In another study, the safety and efficacy of 75% *Morus alba* L. extract oil as a treatment for melasma was evaluated. After 8 weeks of treatment, the melasma area and severity score significantly improved from 4.076 (±0.24) at baseline to 2.884 (±0.25). The results indicate that mulberry extract oil can be an efficacious skin-lightening agent in the treatment of facial melasma [78]. The complex of above-mentioned data indicates that different parts of genus *Morus* plants exhibit excellent tyrosinase activity and are necessary and crucial components for the development of natural and semi-synthetic products for treating hyperpigmentation disorders.

## 4. Conclusions

Herein, the latest advances in searching for potent plant-derived tyrosinase inhibitor compounds isolated from the *Moraceae* family, *Morus* genus and their structure–activity relationship have been discussed (Table 1). The information contained in this article could influence cosmetic and biotechnology companies to take greater interest in considering natural anti-melanogenesis products with satisfactory efficacy and safety for the treatment and cure of hyperpigmentation. Undoubtedly, the major role of structural criterion plays the resorcinol moieties on both rings A and B at positions 2 and 4, and 2′, and 4′, which contribute to enhancing the inhibitory potency. This structural feature should have led to a simplification process of novel formulation design and be treated as a golden rule for the further development of drugs with anti-melanogenesis properties.

The last 20 years have seen milestones not only in the intensive search for the novel formulation of tyrosinase inhibitors, but also for insightful and detailed research into the structural, chemical, crystal, kinetics and mechanisms of inhibition by tyrosinase enzymes. Nevertheless, many aspects of this field remain unresolved. Firstly, most tyrosinase inhibitors have been tested with cheap and commercially available mushroom tyrosinase for use against human tyrosinase [6,7,9]. However, recent reports have indicated significant differences in inhibitor effectiveness between mushroom tyrosinase and human tyrosinase [6,10,17,79]. Secondly, there is still a lack of advanced clinical in vivo trials using *Morus* spp. (as well as focusing on its toxicity, pharmacokinetic and pharmacodynamic), which are crucial for the development of improved clinically potent and compatible tyrosine inhibitors. Most of the research is based on computational or physicochemical models, so the received results must be confirmed in in vivo studies. The majority of constituents tested in silico and/or in physicochemical models will not prove to be beneficial to humans [80]. Thirdly, standardization of the particular extracts from *Morus* spp. would be challenging due to the many varieties and the wide crossing process. Without adequate procedures for authentication and control measures, obtaining reliable and repeatable results of studies in human subjects would be difficult.

## Figures and Tables

**Figure 1 molecules-27-09011-f001:**
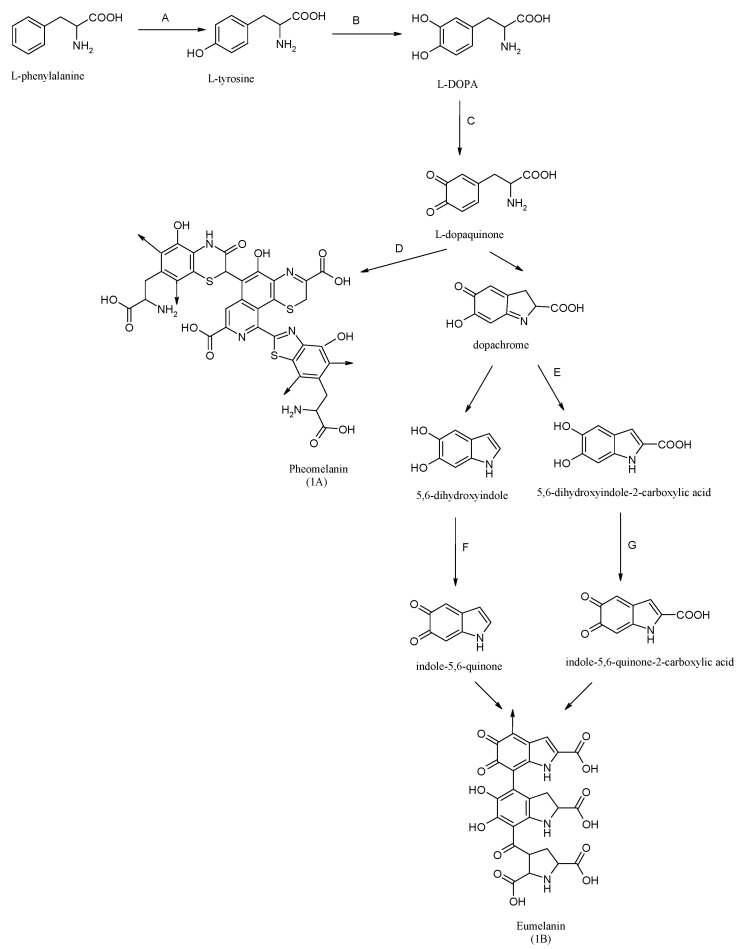
Process of melanogenesis in human epidermal melanocytes. Symbols: A-phenylalanine hydroxylase, B,C,F tyrosinase, D-glutathione or cysteine, E-tyrosinase related protein-2, G-tyrosinase related protein-1. Arrows in (1A) and (1B) mean position of further polymerization of parent molecule.

**Figure 2 molecules-27-09011-f002:**
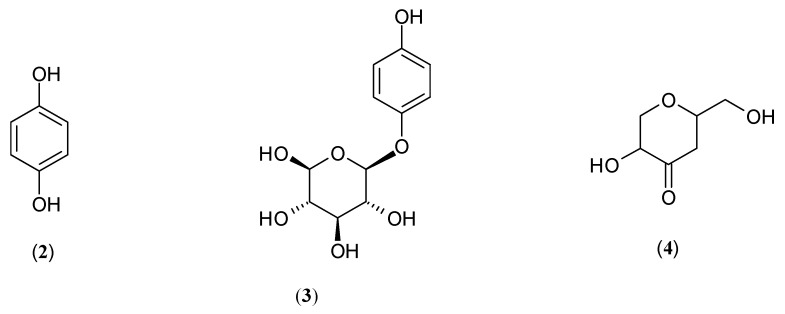
Structures of hydroquinone (**2**), arbutin (**3**) and kojic acid (**4**), frequently used as reference standards.

**Figure 3 molecules-27-09011-f003:**
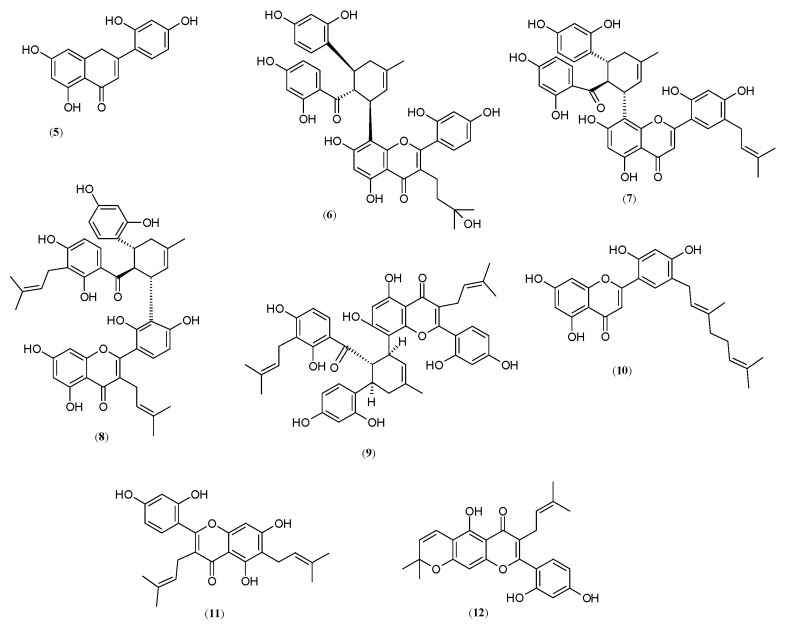
Structures of flavones: norartocarpetin **(5**), moracenin D (**6**), kuwanon G (**7**), kuwanon N (**8**), kuwanon H (**9**), 5′-geranyl-5,7,2′,4′-tetrahydroxyflavone (**10**), cudraflavone (**11**) and cudraflavone B (**12**) with tyrosinase inhibitory activity isolated from genus *Morus*.

**Figure 4 molecules-27-09011-f004:**
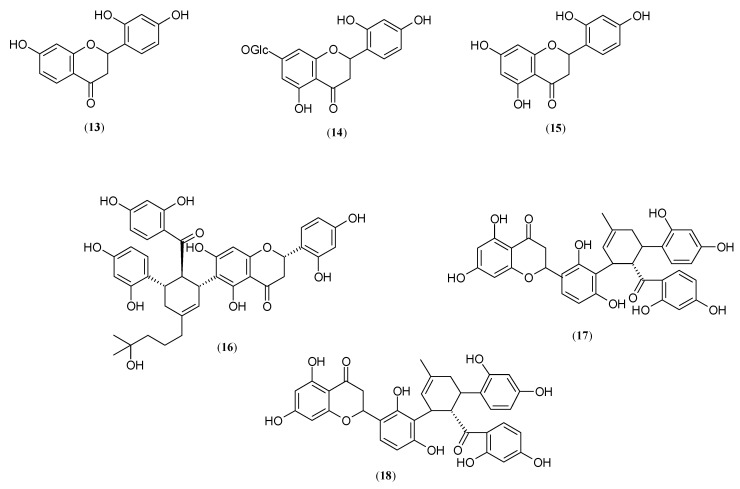
Structures of flavanones and flavonones: 7,2′,4′-trihydroxyflavanone (**13**), steppogenin-7-O-β-D-glucoside (**14**), steppogenin (**15**), sanggenon T (**16**), kuwanon O (**17**) and kuwanon L (**18**) with tyrosinase inhibitory activity isolated from genus *Morus*.

**Figure 5 molecules-27-09011-f005:**
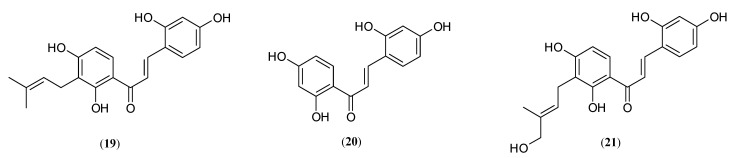
Structures of chalcones: morachalcone A (**19**), 2,4,2′,4′-tetrahydroxychalcone (**20**), 3′-[(E)-4″-hydroxymethyl-2″-butenyl]-2,4,2′,4′-tetrahydroxychalcone (**21**) with tyrosinase inhibitory activity isolated from genus *Morus*.

**Figure 6 molecules-27-09011-f006:**
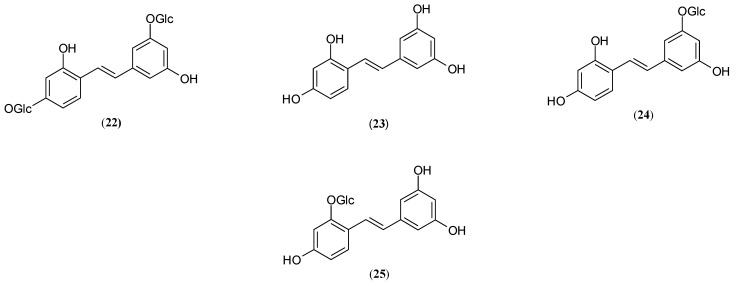
Structures of stilbenes: mulberroside A **(22**), oxyresveratrol (**23**), oxyresveratrol-3′-O-β-D-glucopyranoside (**24**), oxyresveratrol-2-O-β-D-glucopyranoside (**25**) with tyrosinase inhibitory activity isolated from genus *Morus*.

**Figure 7 molecules-27-09011-f007:**
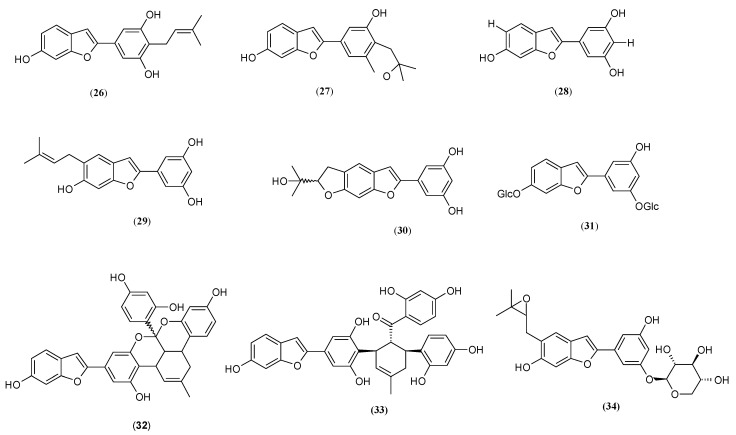
Structures of benzofurans: moracin C (**26**), moracin D (**27**), moracin M (**28**), moracin N (**29**), moracin O (**30**), mulberroside F (**31**), mulberrofuran G (**32**), mulberrofuran J (**33**), moracinoside M (**34**) with tyrosinase inhibitory activity isolated from genus *Morus*.

**Table 1 molecules-27-09011-t001:** Isolated natural compounds with most potent tyrosinase inhibitory activity from genus *Morus*.

Name	Part of Plant	Isolated Compound	IC_50_ (µM)	Relative Inhibitory Strength ^c^	Ref.
*M. alba*	leaf	Norartocarpetin (**5**)	0.0824 ± 0.0089	192.96	[26,58,68]
		Moracin C (**26**)	2.27 ± 0.44	7.0	
		Moracin D (**27**)	12.0 ± 2.5	1.325	
		Moracin N (**29**)	0.924 ± 0.048	17.20	
		Mulberroside F (**31**)	0.29	4.48	
	twig	Steppogenin (**15**)	0.98 ± 0.01	59.49	[43]
		Morachalcone A (**19**)	0.08 ± 0.02	728.75	
		2,4,2′,4′-Tetrahydroxychalcone (**20**)	0.07 ± 0.02	832.86	
		Oxyresveratrol (**23**)	0.10 ± 0.01	583	
		Moracin M (**28**)	8.00 ± 0.22	7.28	
	root	Oxyresveratrol (**23**)	0.49 ^a^, 11.9 ^b^	1503.06 ^d^ 32.60	[53,58]
		Mulberroside A (**22**)	53.6 ± 2.3 ^a^	13.74 ^d^	
*M. australis*	root	Moracenin D (**6**)	4.61 ± 0.13	10.94	[27]
		Sanggenon T (**16**)	1.20 ± 0.04	42.00	
		Kuwanon O (**17**)	1.81 ± 0.08	27.86	
		Oxyresveratrol (**23**)	2.85 ± 0.26	17.69	
		Mulberrofuran G (**32**)	14.65 ± 0.32	3.44	
	stem	Morachalcone A (**19**)	0.82	200.0 ^d^	[50]
		2,4,2′,4′-Tetrahydroxychalcone (**20**)	0.21	780.95 ^d^	
		HMBCH (**21**)	0.17	964.70 ^d^	
*M. bombycis*	cortex	7,2′,4′-Trihydroxyflavanone (**13**)	5.228	21.56	[29]
		2,4,2′,4′-Tetrahydroxychalcone (**20**)	0.9724	115.90	
		Oxyresveratrol (**23**)	3.66	30.80	
		Moracinoside M (**34**)	61.3	1.84	
*M. lhou*	root	Norartocarpetin (**5**)	1.2 ± 0.40 ^a^	13.58	[40]
		Steppogenin (**15**)	1.3 ± 0.30 ^a^	12.53	
		Moracin M (**28**)	7.4 ± 1.0 ^a^	2.20	
	stem bark	Norartocarpetin (**5**)	1.2 ^a^	10.42	[41]
*M. multicaulis*	branchbark	Mulberroside A (**22**)	1.29 ^a^, 7.45 ^b^	nd	[52]
		Oxyresveratrol (**23**)	0.12 ^a^, 0.39 ^b^	nd	
*M. nigra*	root	Kuwanon H (**9**)	10.34 ± 0.19	4.54	[28]
		5′-Geranyl-5,7,2′,4′-tetrahydroxyflavone (**10**)	37.09 ± 1.74	1.26	
		Steppogenin-7-O-β-D-glucoside (**14**)	5.99 ± 0.03	7.84	
		Morachalcone A (**19**)	0.14 ± 0.01	335.36	
		2,4,2′,4′-Tetrahydroxychalcone (**20**)	0.062 ± 0.002	757.26	
		Moracin N (**30**)	30.52 ± 1.46	1.54	
		Oxyresveratrol-3′-O-β-D-glucopyranoside (**24**)	1.64 ± 0.10	28.63	
		Oxyresveratrol-2-O-β-D-glucopyranoside (**25**)	29.75 ± 2.07	1.58	
		Mulberrofuran G (**32**)	17.53 ± 0.26	2.68	
	stem	Morachalcone A (**19**)	0.95 ± 0.040 ^b^	26.19	[49]

^a^ monophenolase used as substrate, ^b^ diphenolase used as substrate, ^c^ compared to kojic acid used as standard reference tyrosinase inhibitor, ^d^ arbutin used as standard tyrosinase inhibitor, nd—no data.

## Data Availability

Not applicable.

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
