# Peer review of "Screening and Structure–Activity Relationship for Selective and Potent Anti-Melanogenesis Agents Derived from Species of Mulberry (Genus Morus)"

_molecules, 2022, doi:10.3390/molecules27249011_

Round 1
Author Response
Dear Reviewer,
Thank you for giving us the opportunity to submit a revised draft of the manuscript entitled „Screening and structure-activity relationship for selective and potent anti-melanogenesis agents derived from species of mulberry (genus Morus)” by Anna Gryn-Rynko, Beata Sperkowska, MichaÅ‚ S.Majewski for publication in Molecules (Manuscript number: molecules-2080608).
We sincerely thank you for your thorough reviews of our manuscript and for the excellent suggestions that we received. We have carefully considered the comments and tried our best to address every one of them. We firmly believe that your comments and suggestions have significantly improved our manuscript. We hope the manuscript after careful revisions meet your high standards. We welcome further constructive comments if any. Below we provide the point-by-point responses.
Sincerely yours,
Anna Gryn-Rynko
Response to Reviewer 1
Comment 1: In general, the manuscript is correct and well-presented but in some part it is hard to understand. Therefore, I recommend the publication after the following major revisions. The topic described in this review is interesting considering the needing for new efficient inhibitors to treat and cure hyperpigmentation. Thus, this review could help researchers focusing in the development of new “druggable” molecules derived from species of mulberry. In some part the review is hard to read and to understand so I suggest the authors to improve the language as I suggested in the comments.
Response: Thank you very much for your nice words about our work. We are very grateful for them. Our manuscript was linguistically rechecked by an English college professor in order to eliminate all inadvertent errors. All grammar, punctuation, spelling, and overall style changes, also suggested by you, were highlighted in the revised manuscript. We tried to removed all grammatical mistakes. Thank you for pointing this out.
Comment 2: I wonder why for the major part of the compounds they report just the monophenolase activity and not the diphenolase one since it is the most described in literature and also faster to evaluate.
Response: Thank you for having raised this important point. I agree with you. Monophenolase activity is more often used. Despite that diphenolase activity is simpler than monophenolase activity since it involves a single catalytic cycle, which rapidly reachesits steady state. Diphenolase activity is also of the Michaelian type and diphenolase activity is generally the most prevalent form of tyrosinase in higher plants. While the monophenolase activity exhibits a lag period τ before reaching steady state (for this reason, adding a certain amount of L-dopa is necessary). I think that there are few reasons. Firstly, kojic acid (the most commonly used commercial inhibitor standard), is reported to be a slow-binding inhibitor of diphenolase activity of tyrosinase (shows a competitive inhibitory effect on monophenolase activity and a mixed inhibitory effect on the diphenolase activity of mushroom tyrosinase). Secondly, not every substrates like catechol and 4-methylcatechol are suitable to define units of diphenolase activity. Generally, the monophenolase activity of tyrosinase is much desired for the industrial synthesis of catechols. Thirdly, diphenolase activity may be reviewed independently, this is not valid for monophenolase activity because chemical reactions of diphenolase and monophenolase production take place simultaneously.

Reviewer 2 Report
1. What is the main question addressed by the research?
In my opinion, the research responds to the need to systematise the scattered knowledge on tyrosinase inhibitors from the genus Morus (family Moraceae). Importantly, these properties relate to the structure of the compounds. 2. Do you consider the topic original or relevant in the field? Does it address a specific gap in the field? 3. What does it add to the subject area compared with other published material? I think the theme is original. It is important and relevant in the fields of health sciences, chemical engineering and the field of pharmaceutical sciences. It systematises knowledge regarding the activity of certain plant derivatives of the genus Morus (Moraceae family) against tyrosinase. The topic is very important for practical applications (for example in cosmetic formulations for the treatment of hyperpigmentary disorders). 4. What specific improvements should the authors consider regarding the methodology? What further controls should be considered? The English language should be improved throughout the text. The manuscript does appear to be methodologically correct. It is coherent and well describes the topics in the theme. I propose to consider the subject further in terms of vitamins A, B3, C and E in mulberry. These compounds also have an action against tyrosinase. The review proposes that the authors answer the questions: Has the subject been considered for the presence of vitamins A, B3, C and E in mulberry. Compounds classified as these vitamins have been shown to exhibit depigmenting effects. Is there any work on this topic? Are the compounds or plant material discussed already of practical application? For example in cosmetics or pharmaceutical products? If so, what properties have been demonstrated for these products? Also depigmentation? Are there studies on this question? 5. Are the conclusions consistent with the evidence and arguments presented and do they address the main question posed? The conclusions are consistent with the evidence and arguments presented. 6. Are the references appropriate? The reference materials are well-selected and up to date, although I am still missing items from 2019-2022. 7. Please include any additional comments on the tables and figures. The tables and figures contain all the necessary information, are appropriately captioned and clear.
Author Response
Dear Reviewer,
Thank you for giving us the opportunity to submit a revised draft of the manuscript entitled „Screening and structure-activity relationship for selective and potent anti-melanogenesis agents derived from species of mulberry (genus Morus)” by Anna Gryn-Rynko, Beata Sperkowska, MichaÅ‚ S.Majewski for publication in Molecules (Manuscript number: molecules-2080608).
We sincerely thank you for your thorough reviews of our manuscript and for the excellent suggestions that we received. We have carefully considered the comments and tried our best to address every one of them. We firmly believe that your comments and suggestions have significantly improved our manuscript. We hope the manuscript after careful revisions meet your high standards. We welcome further constructive comments if any. Below we provide the point-by-point responses.
Sincerely yours,
Anna Gryn-Rynko
Response to Reviewer 2
Comment 1: This manuscript reviews the studies about to updated the recent knowledge about tyrosinase inhibitors derived from of one of the richest in bioactive compounds family Moraceae, genus Morus. In the first section, process of melanogenesis in human epidermal melanocytes are described – this part appears a bit superficial to me, which is probably OK given the focus of the review but the choice of references should be more careful, directing the reader to the most relevant recent papers/reviews. Next section describes polyphenolic compounds – this part provides a very good overview of the subject, here are some minor errors (but irrelevant to the content).
Response: Thank you very much for your nice words about our work. We are very grateful for them. Thank you for remind us that the right choice of literature, especialy the most recent, is essential element determining the quality of publication. That is why some of the publications from the section references have been removed and replaced by the newest one.
Comment 2: It is a nice review. The manuscript is technically correct. The introduction outlines the problem and the research objective. The manuscript is well organized and clearly presents the main results in the related field. In particular, the topic is very important for practical applications (for example in cosmetic formulations for the treatment of hyperpigmentory disorders). Such a review should be helpful to industrial people on how to optimize cosmetic formulation. The reference materials are well-selected and up to date, although I am still missing items from 2019-2022. To sum up, the review is suggested to be published in Molecules. In general, the topic is interesting, the approach is suitable for a this journal.
Response: Thank you very much one more time for your nice words about our work. As it is above-mentioned some of the articles from the section references have been removed and replaced by the newest one from 2019-2022. In addition, we added new section No 3 called Application in cosmetic and pharmaceutical industry. We believe that new adding will improve our manuscript.
Comment 3: But I have the following questions: 1. This is a good overview of literature but I miss a critical insight from the author, because from my point of view, some info (although clearly taken from literature) is misleading.
Response: Thank you for that question. We know that our publication is not perfect and it could be better. In my opinion, in Poland, anti-melanogenesis properties of mulberry are underrated and forgotten. In Poland, there is lack of publications about anti-tyrosinase activity of mulberry as well as products containing mulberry, which can reduce activity of melanogenesis. On the Polish market mulberry plants (especialy M. alba leaves) are sold predominantly as food supplements, which regulate blood glucose metabolism, maintain the recommended blood cholesterol levels and healthy weight, as well as improve cardiovascular health and support immune system. So, we feel that the issue about strong anti-tyrosinase activity of mulberry is extremely important and there is a great need to focus on the above-metioned topic. It is our first work about anti-tyrosinase properties of genus Morus and the more detailed work will be continue.
Comment 4: 2. Has the subject been considered for the presence of vitamins A, B3, C and E in mulberry. Compounds classified as these vitamins have been shown to exhibit depigmenting effects. Is there any work on this topic?
Response: Thank you for that question. Despite that mulberry contains high amount of vitamins (especially vitamin C), which helps reduce excess pigmentation, we have been discussed only isolated and identified compounds from genus Morus. We have not seen publication about antyrosinase effect of isolated vitamins from mulberry. Some of the authors reported the antityrosinase activity of extract from mulberry (new added section No 3 in publication).
Comment 5: Are the compounds or plant material discussed already of practical application? For example in cosmetics or pharmaceutical products? If so, what properties have been demonstrated for these products? Also depigmentation? Are there studies on this question?
Response: As I mentioned before we added new section in our publications No 3 called Application in cosmetic and pharmaceutical industry with few articles about new formulations containing mulberry. In Poland we do not have a lot of cosmetic products containing mulberry. Only few (like serum, creams). They contain white mulberry extract and other plant extract (for example aloe extract, licorice root extract), vitamin C, E. They are reserved for the skin with inflammation, irritation and pigmentation disorders. In Poland, they are not popular.
While, supplements containing white mulberry leaves, are very very common, inexpensive and widely available (pharmacies, herbal shops, grocery stores, gas stations) products. They can be found in different formulations such as capsules, tablets as well as in liquid form, which is currently the most used and popular way of consumption mulberry. They are applied first of all by diabetic and pre-diabetic patients (according to an estimate of International Diabetes Federation in 2021 we have in Poland 2.6 milion people suffering from diabetes and over 1.7 million of adults aged 20–79 years have been undiagnosed with diabetes). There is knowledge deficit about the composition and the amounts of active ingredients in supplements containing mulberry leaves. Concentrations of 1-DNJ and polyphenols, as well as antioxidant potential of mulberry tea, available on the Polish market is unknown.
Comment 5: 4. Page 11, line 388-391: ,,Generally the 2-arylbenzofuran derivatives, obtained form Moraceae hadlower 388 tyrosinase inhibitory properties than the stilbene compounds, suggesting that part of the 389 tyrosinase inhibitory functionality is lost upon formation of the five-membered ring in the 390 2-arylbenzofuran derivatives” - That's why it happens?
Response: Thank you for that question. That is really interesting. Zheng et al. (2010, references No 28) reported that three stilbene glycosides, (such as oxyresveratrol-2-O-β-D-glucopyranoside IC50 =29.75±2.07, oxyresveratrol-30-O-β-D-glucopyranoside IC50 =1.64±0.10, and mulberroside A IC50 >200) have higher antityrosinase activity than 2-arylbenzofuran derivatives (such as moracin M=no data, moracinoside M IC50 >200, moracin O IC50=93.58±0.65, moracin C IC50=111.47±6.96, moracin N IC50=30.5±1.46, mulberrofuran G IC50=17.53±0.26, mulberrofuran J IC50=191.28±4.4, and mulberrofuran B >200). Author suggest that the five-membered ring, that is present in 2-arylbenzofuran derivatives is responsible for lower antityrosinase activity in comparison to stylbenes (there is lack of a five-membered ring in the structure).
Comment 5: 5. The work, in its present form some lacks deeper insight into the relationships between the chemical structure of the compound and the described effects. What I also missed is a more analytical/critical insight into the mechanisms of the interactions described.
Response: Thank you for your valuable feedback. It is really important for us to write a good publication. It is a wonderful direction for us for the next, more detailed publication.

Round 2
Reviewer 1 Report
Dear Authors,
After the modifications you have made your review is now suitable for publication in the present form.